# Low-Cost and Portable Biosensor Based on Monitoring Impedance Changes in Aptamer-Functionalized Nanoporous Anodized Aluminum Oxide Membrane

**DOI:** 10.3390/mi16010035

**Published:** 2024-12-29

**Authors:** Nianyu Jiang, Pranav Shrotriya

**Affiliations:** 1Ames National Laboratory, Mechanical Engineering Department, Iowa State University, Ames, IA 50014, USA; njiang@iastate.edu; 2Mechanical Engineering Department, Iowa State University, Ames, IA 50014, USA

**Keywords:** malaria, amodiaquine sensing, point-of-care sensor, low-cost impedance monitoring, aptamers, point-of-care biosensor

## Abstract

We report a low-cost, portable biosensor composed of an aptamer-functionalized nanoporous anodic aluminum oxide (NAAO) membrane and a commercial microcontroller chip-based impedance reader suitable for electrochemical impedance spectroscopy (EIS)-based sensing. The biosensor consists of two chambers separated by an aptamer-functionalized NAAO membrane, and the impedance reader is utilized to monitor transmembrane impedance changes. The biosensor is utilized to detect amodiaquine molecules using an amodiaquine-binding aptamer (OR7)-functionalized membrane. The aptamer-functionalized membrane is exposed to different concentrations of amodiaquine molecules to characterize the sensitivity of the sensor response. The specificity of the sensor response is characterized by exposure to varying concentrations of chloroquine, which is similar in structure to amodiaquine but does not bind to the OR7 aptamer. A commercial potentiostat is also used to measure the sensor response for amodiaquine and chloroquine. The sensing response measured using both the portable impedance reader and the commercial potentiostat showed a similar dynamic response and detection threshold. The specific and sensitive sensing results for amodiaquine demonstrate the efficacy of the low-cost and portable biosensor.

## 1. Introduction

Electrochemical impedance spectroscopy (EIS) is a well-established method for measuring the dielectric and transport properties of materials, exploring the properties of porous electrodes, and investigating surface reactions [1]. EIS measurements can monitor surface impedance changes in biosensors due to polarity and ionic transport changes associated with the binding of the desired analyte on biorecognition molecule-covered surfaces [2]. Biosensors that rely on EIS have been used for sensing cells [3,4,5], proteins [6,7,8,9], nucleic acids [10,11,12,13], and single molecules [14,15]. EIS measurement also serves as a valuable tool for characterizing surface modification during the immobilization of biomolecules on glass slides [15,16,17], membranes [9], and graphene [18,19,20]. EIS-based investigations have been used to improve sensor performance in wearable sensors [21,22,23] and micro-arrays [21,22,23]. However, a significant limitation of EIS-based measurement-based biosensors is the high cost of commercial electrochemical potentiostats. A low-cost, portable, and easy-to-use impedance reader is required to apply impedance-based biosensors widely.

Several low-cost and portable impedance monitoring devices for biosensing have been reported and rely on either incorporating signal generators with other electronic peripherals [24,25,26,27,28] or commercially developed impedance measurement devices [29]. However, the performance of the impedance monitoring devices has not been compared to commercial potentiostats commonly used for biosensing research. This paper compares the performance of a commercial potentiostat with a low-cost and portable impedance reader based on an analog device-produced microcontroller unit (MCU), ADuCM350. ADuCM350 is an integrated meter-on-chip solution with a microcontroller subsystem for signal processing, control, and connectivity. The low cost (around 20 USD) and small size (1 cm^2^) make ADuCM350 an ideal candidate for a portable impedance monitoring device.

Quinine is found in the bark of trees from the genus Cinchona, and extracts from the bark have been traditionally used in South America to stop shivering, which led to its use as a treatment for malaria [30]. Amodiaquine hydrochloride (AMQ) has a broad therapeutic index [31], and it is a medication used to treat malaria, including uncomplicated plasmodium falciparum malaria [32]. The side effects of AMQ are generally minor to moderate and are like those of chloroquine (CHQ), but in rare cases may lead to liver problems or low blood cell levels [31]. Headaches, trouble seeing, seizures, and cardiac arrest are reported as side effects of AMQ [31]. Additionally, the prevalence of counterfeit malarial medicines due to poor quality control increases the chances of raising drug-resistant diseases and makes the treatment of malaria challenging [33]. Sensors able to accurately sense and confirm the AMQ concentration are required to address these health concerns. Currently, the most common techniques used for sensing AMQ are isothermal titration calorimetry (ITC) [30], poly(vinyl chloride) (PVC) membrane sensors [34], liquid chromatography [35,36,37], conductometry [38], fluorimetry [39,40], electrochemical sensing [41,42,43,44,45,46,47,48], and spectrophotometry [49,50].

However, these techniques are time consuming and require an expensive apparatus and trained operators. A portable, low-cost biosensor that can sense AMQ with a fast turnover time is helpful to address point-of-care sensing needs.

Aptamers are oligonucleotides that are in vitro selected to recognize high affinity and specificity molecular targets and thus may serve as biorecognition molecules for biosensors [51]. A recently reported nucleic-acid-based aptamer (OR7) demonstrates a specific nanomolar affinity binding affinity for AMQ but does not bind specifically to CHQ [30,52]. The OR7 aptamer may be utilized as a biorecognition molecule for AMQ biosensors because of its nanomolar affinity and high specificity for AMQ.

We report a low-cost, portable biosensor composed of an aptamer-functionalized nanoporous anodic aluminum oxide (NAAO) membrane and a commercial microcontroller chip-based impedance reader suitable for electrochemical impedance spectroscopy (EIS)-based sensing. With the sensing substrate assembled inside the electrochemical cell, impedance changes in the aptamer-functionalized membranes on exposure to the target are analyzed by the distribution relaxation time (DRT) method [53,54,55,56]. The impedance measurement and DRT analysis are implemented in the low-cost and portable ADuCM350 MCU-based reader. The performance of the impedance reader is compared to the commercial potentiostat-based sensing results.

## 2. Experiment Section

### 2.1. Reagents and Apparatus

An AMQ-binding aptamer (OR7) was obtained from Integrated DNA Technologies (IDT), with the sequence 5′-C6 Thiol-C AAG GAA AAT CCT TCA ACG AAG TGG G-3′ [30]. Amodiaquine dihydrochloride dihydrate (AMQ) (CAS Number 6398-98-7) and chloroquine diphosphate (CHQ) salt (CAS Number 50-63-5) were purchased from Sigma-Aldrich (St. Louis, MO, USA). The solvent for AMQ was 100% dimethyl sulfoxide (DMSO), purchased from Fisher Scientific (Hampton, NH, USA). All sensing experiments were conducted in phosphate buffer (PBS buffer: 137 mM NaCl, 2.7 mM KCl, 10 mM Ha_2_HPO_4_, 2 mM KH_2_PO_4_, 5 mM MgCl_2_, pH 7.4 at room temperature) with 3% DMSO. All solutions were prepared using double-distilled water (ddH_2_O) produced by the Corning Mega-Pure system (Houston, TX, USA).

NAAO membranes with an average pore diameter of 20 nm and membrane thickness of 50 μm, platinum wires, and silver wires were purchased from Sigma-Aldrich. Silver wires were coated with silver chloride following the previously described procedure to fabricate reference electrodes [9]. An Analog Devices Inc. (Wilmington, MA, USA) evaluation board (EVAL-ADuCM350EBZ) was used to implement the low-cost and portable impedance reader. A commercial potentiostat (Reference 600+ Gamry Instruments Inc. (Warminster, PA, USA)) was also used for the electrochemical experiments.

### 2.2. Transducer Fabrication

NAAO membranes were first cleaned twice with ddH_2_O, ethanol, and isopropanol wash, and a 60 nm thick layer of gold was uniformly coated on the surface of the membrane using a Denton sputter coater. Gold-coated NAAO membranes were washed using ddH_2_O and ethanol twice each before usage. A scanning electron micrograph of gold-coated NAAO membranes with a 60 nm thick gold coating is shown in Figure 1A. The top left of the gold coating in Figure 1A was removed to highlight the contrast between coated and uncoated surfaces. The coating thickness was chosen based on previous work on aptamer-modified sensing experiments [9]. After cleaning, the gold-coated membranes were immersed in 1 µM OR7 aptamer solutions and stored overnight (~17 h) at 4 °C to immobilize the aptamers on the gold surface using thiol–gold (-S-Au-) bonding. Aptamer-coated membranes were immersed in 3 mM mercaptohexane (MCH) for 1 h to displace any physisorbed aptamer from the gold surface to minimize non-specific binding after target introduction. The functionalization steps of the aptasensor are depicted schematically in Figure 1B.

### 2.3. Electrochemical Cell

The aptamer-modified NAAO membrane, Pt wire electrodes, and silver/silver chloride reference electrodes were mounted in the electrochemical cell shown in Figure 1C and exposed to 10 different concentrations of AMQ ranging from 1 to 650 nM. Using a four-electrode arrangement, the potentiostat and ADuCM350-based impedance reader were utilized to monitor the transmembrane impedance change. The aptamer-functionalized membrane is assembled inside two parts of Teflon cells with one working and one reference electrode inserted in each. At the beginning of the sensing experiment, both sides of the electrochemical cell were filled with a 500 μL PBS buffer as the electrolyte. Working and counter electrodes were inserted inside each chamber to measure the current flow through the NAAO membrane. Two reference electrodes were inserted into each chamber to maintain the desired alternative voltage value. AMQ was injected into the half-cell at the gold-coated side. In addition, impedance changes due to membrane exposure to different concentrations of chloroquine molecules that do not specifically bind to the AMQ-binding aptamer were also measured as a control response.

For the sensing experiments with the potentiostat, the four electrodes in the cells were connected to the corresponding ports, and EIS scans were conducted to monitor the impedance changes across the membranes with exposure to 10 different AMQ concentrations. The frequency range of voltage excitation was set between 0.1 and 100,000 Hz with 10 points for each decade. Five signal periods were performed at each frequency excitation to minimize any possible signal aliasing. At each AMQ titration, three cycles of the frequency scan were conducted to ensure the molecule had enough time to diffuse through the electrolyte. Signals at all cycles were overlaid and observed to ensure a steady state was reached. Each titration took twenty minutes, and the overall EIS measurement was finished within three hours.

For the sensing experiments with the impedance reader, the connections between the impedance reader and the electrochemical cell are shown in Figure 2. Pins #3 and #6 on the EV-ADUCM350-BIO3Z were registered as working and counter electrodes, while pins #4 and #5 were registered as reference electrodes. An alternating voltage was applied through pins #4 and #5, and the corresponding alternating current was measured through pins #3 and #6. The chip Cortex-3 generated the alternating voltage perturbation required for EIS, and an FFT (fast Fourier transformation) was used to perform the data analysis. The alternating voltage perturbation excitation frequency was varied between 300 and 10,000 Hz. The electrochemical current corresponding to the applied excitation voltage was measured and converted to a voltage by an Op-Amplifier. The magnitude and phase data were calculated by comparing the signal from the biosensor to a resistor printed on ADuCM350 with a fixed resistance. The signal measured by the impedance reader was transferred to the laptop using an emulator board with a serial port.

Experimentally measured EIS responses from both the impedance reader and the potentiostat were analyzed using the distribution of relaxation time (DRT) approach [53,55,57] using the parameters listed in Appendix A.

## 3. Result and Discussion

### 3.1. Impedance Changes in the Membrane Exposed to AMQ

Typical Bode plots (magnitude and phase components) of the impedance of the OR7 aptamer-functionalized membrane before and after exposure to 250 nM of AMQ obtained using a commercial potentiostat are plotted in Figure 3A. The Bode plots obtained using the impedance reader for a similar change in AMQ concentration on the aptamer-functionalized membrane are plotted in Figure 3B. The Bode plots obtained using both the potentiostat and impedance reader show a similar magnitude of impedance and an increase in the membrane impedance on exposure to AMQ. The frequency range of the impedance reader measurements is limited compared to the Gamry Potentiostat. Still, over the measured frequency ranges, both the instruments show a similar magnitude and phase of membrane impedances.

The Bode plots can provide a qualitative comparison of the impedance changes over the measurement frequency range but cannot give a quantitative measure for the sensing response. The measured impedance data were fitted to the distribution of relaxation times (DRT) shown in Equation (1) [53,58]
(1)$$Z\left(f\right)=G_{\infty }+\int_{-\infty }^{\infty }\frac{G\left(\tau \right)}{1+i2\pi f\tau },$$
where *f* is the frequency, *G_∞_* is the solution resistance, and *G*(*τ*) is the resistance corresponding to the relaxation time, *τ.* The deconvolution of the impedance is conducted to identify the relaxation times that are most sensitive to AMQ/aptamer binding on the membrane. The DRT spectrum [56] corresponding to impedances plotted in Figure 3A,B are plotted in Figure 3C,D, respectively. The DRT plots in Figure 3A,B indicate that the magnitude of the resistance has a peak, corresponding to the relaxation time of approximately 0.01 s. The magnitude of the peak in the DRT spectrum increases on the exposure of the membrane to a 250 nM AMQ concentration for both Figure 3C,D. Hence, the sensing signal was defined according to Equation (2)
(2)$$Signal\left[\frac{∆R}{R}\right]=\frac{G_{n}\left(\tau_{p}\right)-G_{0}\left(\tau_{p}\right)}{G_{0}\left(\tau_{p}\right)}$$
where $G_{n}\left(\tau_{p}\right)$ and $G_{o}\left(\tau_{p}\right)$ are the magnitudes of the DRT peak for the membrane exposed to *n* and a zero concentration of AMQ and $\tau_{p}$ is the relaxation time corresponding to the DRT peak. The sensing signal is normalized to ensure that the variations in impedance changes across different membranes do not influence the sensing signal.

The sensing signals corresponding to the exposure of the OR7 aptamer-functionalized membrane to at least 10 different concentrations of AMQ measured using the potentiostat and impedance reader are plotted in Figure 4A,B, respectively. The OR7 aptamer binds specifically to AMQ but does not have binding affinity to CHQ molecules [30,52], so CHQ serves as a valuable control for the biosensor response. The sensing signals obtained by the exposure of the aptamer-functionalized membranes to chloroquine (CHQ) are also plotted in Figure 4A,B, respectively. The sensing signal increases with the increase in AMQ concentrations, but the sensing signals are not correlated to changes in CHQ concentrations. The sensing signal generated from the control experiment (CHQ) is between −5% and 0% and is separated from AMQ sensing data. The biosensor-specific response to AMQ over CHQ in Figure 4A,B may be attributed to the specific binding between the OR7 aptamer and AMQ molecules on the sensing surface. The sensing and control signals measured using the potentiostat (Figure 4A) and low-cost impedance reader (Figure 4B) are of similar magnitude and show similar trends with changes in AMQ and CHQ concentrations.

The sensing responses corresponding to different concentrations were fitted into the Langmuir isotherm shown in Equation (3)
(3)$$\frac{∆R}{R}=F_{max}\frac{C}{C+K_{D}}$$
where *F_max_* is the maximum sensing response, *C* is the concentration of AMQ, and *K_D_* is the dissociation constant for binding between the OR7 aptamer and AMQ molecules. The fits into the sensing responses and the 95% confidence interval of the fits are plotted along with data measured using the potentiostat and impedance reader in Figure 4A,B, respectively. In both figures, the Langmuir fit gives a *F_max_* value of 10%, indicating that the sensing signal saturates at around 10% with the increase in AMQ concentrations. The dissociation constant (*K_D_*) values are 2 nM from potentiostat data and 4 nM from impedance reader data.

The dynamic range of the AMQ sensing experiment using the potentiostat and impedance reader are shown in Figure 4C,D, respectively. The goodness of fit of linear regression in the dynamic range plots is 96.9% and 99% for data measured by the potentiostat and impedance reader. Using the criteria 3.3S_y_/S, where S_y_ represents the standard deviation and S represents the slope of the curve, the limit of detection (LOD) for the potentiostat is estimated to be 10 nM, while for the impedance reader, it is estimated to be 25 nM.

The sensing response or the change in transmembrane impedance depends on the influence of the aptamer/AMQ binding on the transport of ions through the nanoporous structure of NAAO. Steric hindrance and surface charge distribution changes are the dominating mechanisms underlying the sensing response for pore-based biosensors. The average pore size of the NAAO membrane is 20 nm, based on the SEM images. The aptamer has 26 bases and is approximately 9 nm in diameter. Since AMQ’s size is small, the OR7/AMQ complex is expected to be the same size as the aptamer and smaller than the NAAO pore size. The small size change in the unbound aptamer and bound aptamer/AMQ complex is unlikely to cause a significant change in the steric hindrance to ionic transport through the pore. However, the change in surface charge distribution may result in ionic conductance changes through the pores [59]. The ionic conductance through the pore may be separated into surface-charge-governed or geometry-governed regimes, depending on the value of the non-dimensional constant ($\frac{\sigma }{neh}$), where *σ* is surface charge density, *n* is electrolyte concentration, *e* is the elementary charge, and *h* is the channel diameter. The conductance is governed when the non-dimensional ratio of the surface charge distribution to electrolyte concentration is less than 1. In the current experiments, the surface charge density, *σ*, due to the self-assembled monolayer of the OR7 aptamer, is about −50 mC/m^2^ [60], *n* is 0.15 M, and *h* is 20 nm for the alumina membrane. Thus, ionic conductance through the aptamer-covered pores is dominated by the surface charge distribution. The pK_a_ of AMQ in PBS is between 9 and 10, which means that AMQ is positively charged in the PBS buffer (pH~7.0). Binding between AMQ and the aptamer-covered negatively charged surface will likely result in quenching of the negative surface charge and thus decrease ionic conductance through the pores, as schematically represented in Figure 5A. The increase in AMQ concentration results in the increased attenuation of ionic conductance and higher impedance across the membrane. The CHQ molecules with a pKa of 10.1 are also positively charged in the PBS buffer. However, the non-specific interaction between CHQ and the aptamer-functionalized membrane results in minimal changes in the surface charge distribution and ionic transport, as shown schematically in Figure 5B.


### 3.2. Impedance Reader

The dynamic range and LOD for AMQ detection reported in the literature are compared to current measurements in Table 1(A), and it shows that current aptasensor-based measurements can achieve a low LOD. Still, the dynamic range is limited due to the high affinity of the OR7 aptamer for AMQ molecules. The performance of the low-cost impedance reader is compared to other low-cost biosensors reported in the literature. The performance of the biosensor is comparable to the other low-cost sensing devices and provides a proof of concept for implementing a low-cost, portable impedance reader that may be used to develop point-of-care diagnostic devices.

## 4. Conclusions

We report a low-cost, portable biosensor comprising an aptamer-functionalized NAAO membrane and a commercial microprocessor-based impedance reader. The biosensor performance is characterized by sensing AMQ using a specifically binding aptamer (OR7) as a biorecognition element. The low-cost biosensor provides specific detection of AMQ with an LOD of 25 nM and a dynamic range of 1–1000 nM. A commercial potentiostat used with aptamer-functionalized membranes for AMQ sensing provides an LOD of 10 nM and a dynamic range of 1–1000 nM, indicating that the low-cost impedance reader can replicate the sensing performance of a commercial potentiostat. The impedance of the aptamer-functionalized membrane increases in the presence of specifically binding AMQ but does not change in the presence of CHQ, indicating that the response is due to the specific aptamer/AMQ binding on the sensor surface. The sensor performance is comparable to other low-cost and portable sensors and can provide the framework for implementing point-of-care diagnostic devices.

## Figures and Tables

**Figure 1 micromachines-16-00035-f001:**
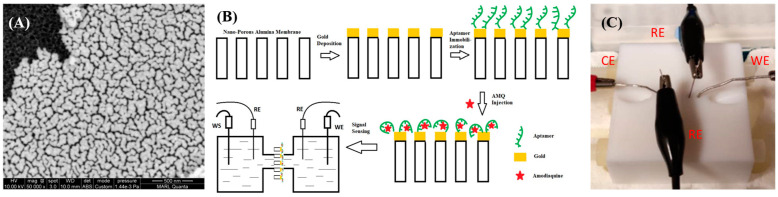
(**A**) SEM image of the gold-coated membrane, (**B**) Teflon cells for electrochemical experiments and the four-electrode setup, (**C**) sensor structure with four electrodes (RE—reference electrode, WE—working electrode, CE—counter electrode) during the process of aptamer immobilization and the impedance measurement.

**Figure 2 micromachines-16-00035-f002:**
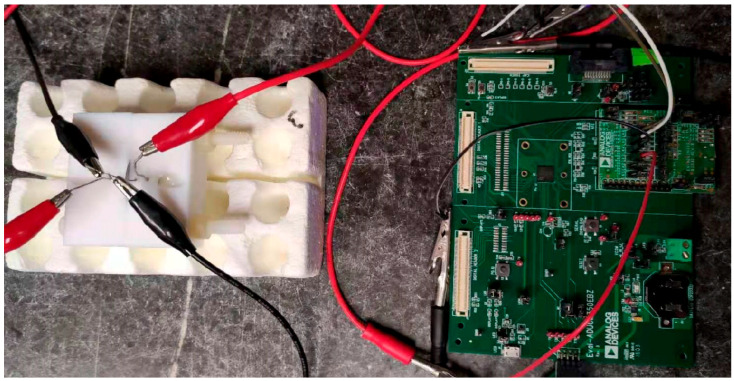
Evaluation kit of the ADuCM350-based impedance reader and the connection between the reader and the electrochemical cell.

**Figure 3 micromachines-16-00035-f003:**
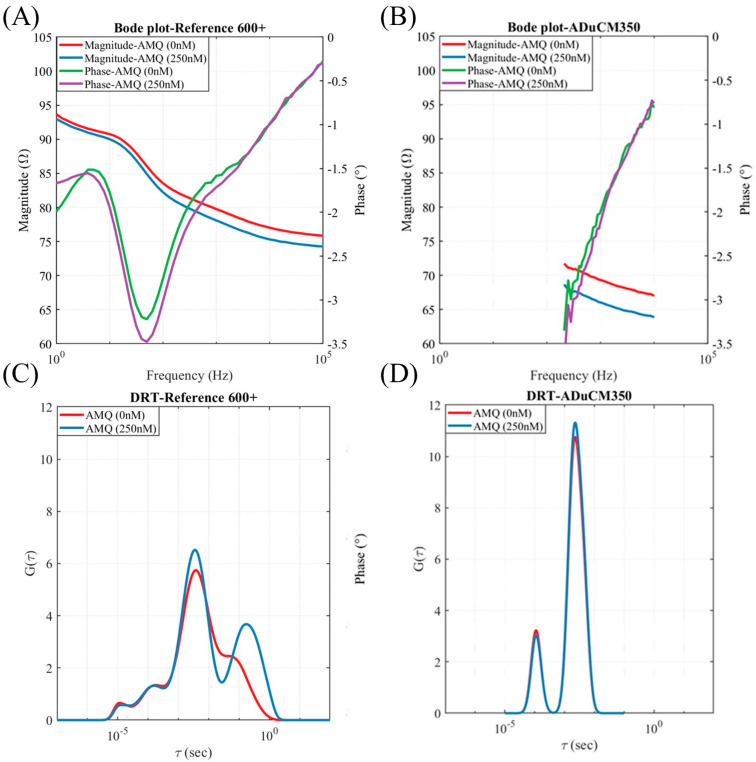
Impedance of the OR7-functionalized membrane exposed to no AMQ and 250 nM AMQ inside the electrochemical cell: (**A**) Bode plot measured using a commercial potentiostat, (**B**) Bode plot measured using a low-cost impedance reader, (**C**) DRT deconvolution of the Bode plot in (**A**), and (**D**) DRT deconvolution of the Bode plot in (**B**).

**Figure 4 micromachines-16-00035-f004:**
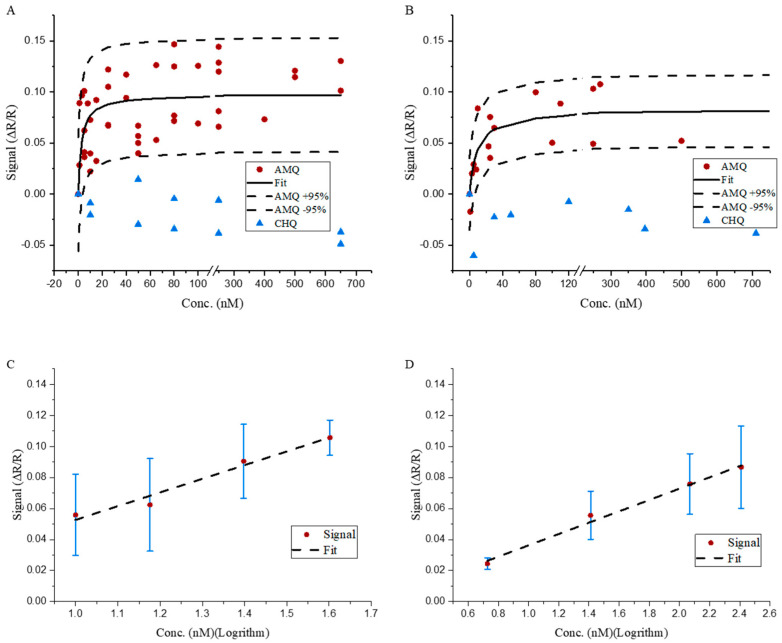
Sensing result of AMQ using both (**A**) potentiostat reference 600+ and (**B**) the ADuCM350-based impedance reader. AMQ LOD uses (**C**) potentiostat 600+ and (**D**) the ADuCM350-based impedance reader.

**Figure 5 micromachines-16-00035-f005:**
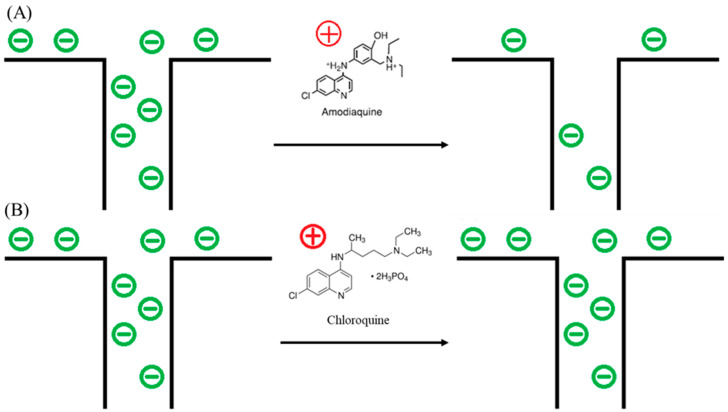
(**A**) Quenching mechanism of AMQ binding with OR7, (**B**) charge accumulation of CHQ physisorption onto NAAO membrane surface. (⊝ indicates surface negative charge and ⊕ indicates positive charge of AMQ and CHQ).

**Table 1 micromachines-16-00035-t001:** (**A**) Recent publications about AMQ biosensors. (**B**) Recent publications about portable biosensors.

(**A**)			
**Publications**	**Detection Method**	**Dynamic Range**	**Limit of Detection**
Chiwunze et al. [44]	Electrochemical	100–3500 nM	89 nM
Malongo et al. [34]	Electrochemical	10–3200 nM	1200 nM
Karakaya et al. [43]	Electrochemical	500–25,000 nM	160 nM
Valente et al. [61]	Electrochemical	x	9272 nM
Potentiostat-Based Measurement	Electrochemical	1–1000 nM	10 nM
Impedance Reader-Based Measurement	Electrochemical	1–1000 nM	25 nM
(**B**)			
**Publications**	**Targets**	**Dynamic Range**	**Limit of Detection**
Nagabooshanam et al. [62]	chlorpyrifos	10 to 100 ng/L	6 ng/L
Sawhney and Conlan [26]	cancer antigen 125	0.92–15.20 ng/μL	0.24 ng/μL
Gupta et al. [63]	Pb(II)	0.001–1000 nM	0.81 nM
This Work	AMQ	1–1000 nM	25 nM

## Data Availability

The original contributions presented in the study are included in the article, further inquiries can be directed to the corresponding author.

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
