# Peer review of "Low-Cost and Portable Biosensor Based on Monitoring Impedance Changes in Aptamer-Functionalized Nanoporous Anodized Aluminum Oxide Membrane"

_micromachines, 2024, doi:10.3390/mi16010035_

Round 1

Reviewer 1 Report

Comments and Suggestions for Authors

1. As is known, commercial potentiostats are relatively inexpensive devices. What is the commercial benefit of using the approach developed by the authors? Is there a need to produce cheaper devices?

2. For what practical problems is the developed Portable Biosensor needed? Where is it necessary to control the considered analytes using the Portable Biosensor at the sampling site? It is necessary to clarify these points in the manuscript.

3. The literature review contains few recent publications; it is necessary to refer mainly to developments of the last 5 years.

Author Response

  1. As is known, commercial potentiostats are relatively inexpensive devices. What is the commercial benefit of using the approach developed by the authors? Is there a need to produce cheaper devices?

The cost of commercial electrochemical potentiostats capable of electrochemical impedance spectrometry (EIS) measurements varies from $1000 - 20000 dollars. This cost makes the deployment of EIS-based sensors difficult in resource-limited countries as well as for point-of-care devices. A low-cost and portable potentiostat is required to implement sensors for point-of-care and sensing applications.

  1. For what practical problems is the developed Portable Biosensor needed? Where is it necessary to control the considered analytes using the Portable Biosensor at the sampling site? It is necessary to clarify these points in the manuscript.

The following texts have been included to explain the importance of the portable biosensors, especially malaria durg sensors:

EIS-based investigations have been used to improve sensor performance in wearable sensors (21–23) and micro-arrays (21–23). However, a significant limitation of EIS-based measurements-based biosensors is the high cost of commercial electrochemical potentiostats. A low-cost, portable, and easy-to-use impedance reader is required to apply impedance-based biosensors widely.

Amodiaquine hydrochloride (AMQ) has a broad therapeutic index(31), and it is a medication used to treat malaria, including uncomplicated Plasmodium falciparum malaria (32). The side effects of AMQ are generally minor to moderate and are like those of Chloroquine (CHQ), but in rare cases may lead to liver problems or low blood cell levels (31). Headaches, trouble seeing, seizures, and cardiac arrest are reported as side effects of the AMQ(31). Additionally, the prevalence of counterfeit malarial medicines due to poor quality control increases the chances of raising drug resistant disease and the treatment of Malaria challenging (33). Sensors able to accurately sense and confirm the AMQ concentration are required to address these health concerns            

  1. The literature review contains few recent publications; it is necessary to refer mainly to developments of the last 5 years.

We have updated the literature review to include recent developments in Amodiaquine (AMQ) sensing and a review article on the importance of AMQ sensing.

Additionally, the prevalence of counterfeit malarial medicines due to poor quality control increases the chances of raising drug resistant disease and the treatment of Malaria challenging (33). Sensors able to accurately sense and confirm the AMQ concentration are required to address these health concerns. Currently, the most common techniques used for sensing AMQ are Isothermal titration calorimetry (ITC)(30), poly(vinyl chloride) (PVC) membrane sensor (34), liquid chromatography(35–37), conductometry(38), fluorimetry(39,40), electrochemical sensing (41–48) and spectrophotometry(49), (50).

Reviewer 2 Report

Comments and Suggestions for Authors

The article titled “Low-Cost and Portable Biosensor Based on Monitoring Impedance Changes in Aptamer Functionalized Nanoporous Anodized Aluminum Oxide Membrane” reports a low-cost, portable biosensor composed of aptamer functionalized nanoporous anodic aluminum oxide (NAAO) membrane and a commercial microcontroller chip-based impedance reader suitable for electrochemical impedance spectroscopy (EIS) based sensing. There are some places where improvement is required. This article can be published after addressing the following comments.

1.       Authors are asserting a comparable dynamic response between the low-cost impedance reader and the commercial potentiostat but provide only minimal quantitative validation or statistical significance to support this claim.

2.       Also, for selectivity studies, authors have ineffectively explained experimental control procedures applied to moderate influence from non-target molecules, aside from chloroquine. In my opinion some other molecules should be tesetd besides chloroquine.

3.       In addition to this, I am not satisfied with the explanation of the aptamer functionalized sensor fabrication (e.g., coating thickness fluctuates without justification).

4.       Moreover, is it possible to add at least 3 different concentrations of AMQ (only 250 nM is not sufficient to justify the claim)?

Comments on the Quality of English Language

Check for syntax errors

Author Response

  1. Authors are asserting a comparable dynamic response between the low-cost impedance reader and the commercial potentiostat but provide only minimal quantitative validation or statistical significance to support this claim.

Detailed statistical and comparison is presented in the following paragraphs in the manuscript:

In both Figures, the Langmuir fit gives a Fmax value of 10%, indicating that the sensing signal saturates at around 10% with the increase in AMQ concentrations. The dissociation constant (KD) values are 2nM from potentiostat data and 4nM from impedance reader data.

The dynamic range of the AMQ sensing experiment using potentiostat and impedance reader are shown in Figures 4C and 4D, respectively. The goodness-of-fit of linear regression in the dynamic range plots is 96.9% and 99% for data measured by potentiostat and impedance reader. Using the criteria 3.3Sy/S, where Sy represents the standard deviation, and S represents the slope of the curve, the limit of detection (LOD) for the potentiostat is estimated to be  10nM, while for the impedance reader is estimated to be 25nM.

  1. Also, for selectivity studies, authors have ineffectively explained experimental control procedures applied to moderate influence from non-target molecules, aside from chloroquine. In my opinion some other molecules should be tesetd besides chloroquine.

We agree that for an ideal selectivity study the sensor should be tested against a range of different molecules. We are currently working on these experiments. However, this paper focuses on reporting a low-cost sensor and comparing the performance of a low-cost impedance reader and a commercial potentiostat. We have compared the sensor response between amodiaquine and chloroquine because of the similairy in their chemical structures:

  1. In addition to this, I am not satisfied with the explanation of the aptamer functionalized sensor fabrication (e.g., coating thickness fluctuates without justification).

We thank the reviewer for catching the inadvertent mismatch in the coating thickness mentioned in the manuscript. The transducer was coated with a gold film of 60 nm thickness and the manuscript has been updated.

  1. Moreover, is it possible to add at least 3 different concentrations of AMQ (only 250 nM is not sufficient to justify the claim)?

The sensor response on exposure to at least 10 different concentrations of AMQ and CHQ to characterize the response is shown in Figs 4A and 4B, respectively.  The detailed analysis for sensor response to 250 nM concentration is presented to explain the data processing used to obtain the sensor response.

Round 2

Reviewer 1 Report

Comments and Suggestions for Authors

I recommend for publication